# Eating Competent Parents of 4th Grade Youth from a Predominantly Non-Hispanic White Sample Demonstrate More Healthful Eating Behaviors than Non-Eating Competent Parents

**DOI:** 10.3390/nu11071501

**Published:** 2019-06-30

**Authors:** Barbara Lohse, Melissa Pflugh Prescott, Leslie Cunningham-Sabo

**Affiliations:** 1Rochester Institute of Technology Wegmans School of Health and Nutrition, Rochester, NY 14623, USA; 2Department of Food Science and Human Nutrition, University of Illinois Urbana-Champagne, Urbana, IL 61801, USA; 3Department of Food Science and Human Nutrition, Colorado State University, Fort Collins, CO 80523, USA

**Keywords:** eating competence, parent feeding behavior, school-age youth, Hispanic, replication

## Abstract

The purpose of this study was to determine if the associations between eating competence (EC) and eating behaviors that were found in a USA sample of predominantly Hispanic parents of 4th grade youth could be replicated in a USA sample of predominantly non-Hispanic white parents of 4th graders. Baseline responses from parents (*n* = 424; 94% white) of youth participating in a year-long educational intervention were collected using an online survey. Validated measures included the Satter Eating Competence Inventory (ecSI 2.0^TM^), in-home fruit/vegetable (FV) availability, healthful eating behavior modeling, and FV self-efficacy/outcome expectancies (SE/OE). Data were analyzed with general linear modeling and cluster analyses. The findings replicated those from the primarily Hispanic sample. Of the 408 completing all ecSI 2.0^TM^ items, 86% were female, 65% had a 4-year degree or higher, and 53% were EC (ecSI 2.0^TM^ score ≥ 32). Compared with non-EC parents, EC modeled more healthful eating, higher FV SE/OE, and more in-home FV availability. Behaviors clustered into those striving toward more healthful practices (*strivers*; *n* = 151) and those achieving them (*thrivers*; *n* = 255). *Striver* ecSI 2.0^TM^ scores were lower than those of *thrivers* (29.6 ± 7.8 vs. 33.7 ± 7.6; *p* < 0.001). More EC parents demonstrated eating behaviors associated with childhood obesity prevention than non-EC parents, encouraging education that fosters parent EC, especially in tandem with youth nutrition education.

## 1. Introduction

Parents and caregivers inform children’s food and eating habits by shaping home and social environments and through actions (intended or not) that construct attitudes and beliefs [1,2]. Parent practices and attitudes associated with healthful eating and making healthy foods more available help promote intake by children [2,3]. One approach to eating that is strongly associated with these healthful practices is eating competence (EC). EC, characterized as an intra-individual approach to eating and food-related attitudes and behaviors that entrains positive biopsychosocial outcomes [4,5], has been associated with higher diet quality [6,7,8,9], more healthful eating behaviors [8,10,11,12,13] and other lifestyle practices (e.g., being more physically active) [14], better sleep hygiene [15], and food resource management skills [12,13]. Competent eaters give themselves permission to eat foods they like in amounts that are satisfying, but have the discipline to provide nourishing meals and snacks on a routine basis. They enjoy eating and do not feel guilty about this enjoyment, experiment with and eat a variety of foods, plan meals, make time to eat, and shape an environment that helps them attend to eating [4].

In a sample of parents of 4th graders (*n* = 339), which was mostly Hispanic (78%) with 35% denoting Spanish as their primary language, the parents who were EC demonstrated more modeling of regular meals and fruit and vegetable (FV) intake, better parent self-efficacy for providing meals and snacks that include FV, and greater FV availability in the home [5]. Parents denoted by cluster analysis as *achievers* (i.e., they more frequently ate dinner and breakfast with their child; had higher scores on scales that measured healthful modeling behavior, self-efficacy, and availability) were also more EC than the *strivers*, who demonstrated fewer of these desired behaviors. *Achiever* EC score was 34.9 ± 7.8 versus the *Striver* score of 30.3 ± 8.9, *p* < 0.001) [5]. The ethnicity of the sample is important because Hispanic children have a higher prevalence of being overweight or obesity [16,17,18], and Hispanic parents underestimate their child’s weight status [19]. In Colorado, Hispanic children have nearly twice the rate of obesity as non-Hispanic white youth [20]. In addition, Hispanic parent feeding behaviors and attitudes may differ from non-Hispanics in ways that conflict with obesity prevention recommendations [21,22,23,24,25,26], for example, a low demanding/high responsive feeding style [24], setting limits on what and how much their elementary school children ate, punishing a child who did not want to eat [23], or using high levels of restriction and pressure to eat [21,26]. Comparing Hispanic parents to African–American parents using video capture strategies, Hughes et al. [22] noted high levels of Hispanic parental encouragement to eat (which puts their children at risk for overweight), but reduced use of high control strategies for their obese children. Child obesity prevention messages were interpreted differently among Hispanic, white, and black mothers of children 3 to 10 years old [25]. Hispanic mothers did not connect messages about calories, eating more FV, or family meals to impact on child weight and, unlike non-Hispanic white mothers, Hispanic mothers catered to a child’s preference more frequently. 

To date, conclusions about the relationship of EC to parent eating behaviors is limited to the sample of Hispanic parents [5]; generalization to other ethnicities has not been investigated. The importance of reproducibility of the survey results is not a new phenomenon, but the call to attend to it has become urgent in psychology and the social sciences [27,28,29,30,31,32]. Concerns about these external validation issues align with recent calls to consider reproducibility in behaviorally focused studies [28,33], by encouraging replication [27] and especially publication of replication research across study groups [29]. Therefore, the purpose of this study was to determine if the relationships previously reported between parent eating and feeding behaviors and EC in a sample of predominantly Hispanic parents participating in a school and family-based nutrition education intervention [5] are replicated in a sample of predominately non-Hispanic white parents participating in the same intervention. Findings will inform the external validity of the associations between EC and parent eating and feeding behaviors.

## 2. Materials and Methods

### 2.1. Study Design and Participants

This descriptive, cross-sectional study utilized baseline data from parents with 4th grade children enrolled in schools participating in the controlled, school level randomized trial of *Fuel for Fun’s (FFF)* [34] impact on FV preferences, attitudes, and self-efficacy toward cooking and making foods in 83 classrooms across eight schools in Fort Collins, Colorado from 2012–2016. Study design and methods paralleled those utilized for study of the intervention in the predominantly Hispanic sample [5]. Institutional review boards at the Blinded for Review Universities approved this project; CSU#12-3278H. The study registration at ClinicalTrials.gov is 5NCT02491294.

### 2.2. Data Collection 

Parents of 4th grade youth from the participating schools were recruited uniformly using flyers brought home by children and school e-blasts that invited them to visit a website to read about the study. Eligible participants (i.e., they could read English and were parents or the primary caregiver for a 4th grader in the participating schools) were directed to the informed consent, then clicked the box denoting agreement to participate. Phone and email contacts of study personnel were included in the consent document. Printing the consent was a suggested action. Consented parents were directed to the online survey (Qualtrics, Provo, UT, USA). Classroom rosters were compared against survey entries of child name, teacher, and school to assure that the survey responder was eligible. After eligibility was affirmed, responders were sent an e-gift card for a discount or online store. Reminders to parents to access the study were sent 2–4 weeks after the initial announcement and enrollment was closed just prior to the start of the classroom intervention. 

### 2.3. Measures

#### 2.3.1. Modeling 

The online instrument included a tested, face valid scale with internal consistency that assessed the frequency of 11 modeling behaviors related to meals and FV over the past week, with four response options ranging from never (0 days) to always (7 days) [35,36]. Possible scores ranged from 0 to 33. Cronbach’s alpha was 0.78 for modeling, suggesting internal consistency.

#### 2.3.2. Self-Efficacy and Outcome Expectancy

Perception of self-efficacy and outcome expectancies (SE/OE) related to preparing and offering FV that would be accepted by their children was assessed using a validated 12-item scale with five response options ranging from strongly disagree to strongly agree [37]. Scores could range from 12 to 60. Cronbach’s alpha was 0.95 for SE/OE. 

#### 2.3.3. Fruit and Vegetable Availability

In-home FV availability over the past week was determined by previously tested parental report of the presence or absence of eight fruits, nine vegetables, and three types of 100% juice in fresh, frozen, canned, or dried form [38,39]. 

#### 2.3.4. Eating Competence

EC was measured using the construct validated Satter Eating Competence Inventory (ecSI 2.0^TM^) [12], which consists of 16 items scored from 0 to 3 (never/rarely to always). ecSI 2.0^TM^ scores can range from 0 to 48, and can be categorized into 4 ecSI 2.0^TM^ subscales: internal regulation, food acceptance, eating attitudes, and contextual skills, with possible subscale scores ranging from 0 to 9 (internal regulation, food acceptance) or 0 to 15 (eating attitudes, contextual skills). ecSI 2.0^TM^ scores of 32 or greater denote EC. Cronbach’s alpha was 0.90 for ecSI 2.0^TM^.

#### 2.3.5. Physical Activity

Physical activity of parents was assessed using the validated International Physical Activity Questionnaire (IPAQ), which measures physical activities over the previous seven days. Responses were converted to metabolic equivalent of task (MET) minutes/week and used to determine categories of low, moderate, and vigorous activity [40]. Weight and height were self-reported. 

#### 2.3.6. Caregiver Feeding Styles

Parent feeding style was assessed using the Caregiver’s Feeding Style Questionnaire [22], a validated measure that consists of 19 items with five response options. Scores are converted to four caregiver feeding styles: uninvolved, indulgent, authoritarian and authoritative. In this sample, Cronbach’s alpha was 0.88.

#### 2.3.7. Stress

To measure parent stress using a validated measure, but with minimal respondent burden, stress was measured using the single item included in the Community Health Database of Southeast Pennsylvania [41], which has been administered every two to three years since 1991. Littman et al. [42] demonstrated that measuring stress with a single item was a reliable, valid, and practical choice to limit assessment length.

#### 2.3.8. Income

On the basis of recommendations made with the previous Hispanic sample [5], a proxy measure of parent income, previously tested for face and content validity, was added to the online survey [43]. Parents were considered low-income if they used ≥1 income-based assistance program or always or often worried about money for food. 

### 2.4. Statistical Analysis 

Data were analyzed with SPSS (version 25.0, 2017, Chicago, IL, USA). For each scale, item responses were summed to create a scale score with higher scores on the parent scales indicating preferred practice; missing items abrogated a total scale score. Parent modeling, SE/OE, and in-home FV availability results were categorized as below the median and at or above the median. EC was defined as ecSI/LI score ≥ 32 [12]. 

#### 2.4.1. Psychometrics 

Internal consistency for all surveys was affirmed with Cronbach’s alpha > 0.77; EC score distribution was affirmed as normal with no kurtosis, absolute skewness < 0.25, and a straight line Q–Q plot. Individual SE/OE items were compared between EC and non-EC parents with a Mann–Whitney *U* test, because they were not normally distributed. Total SE/OE score was transformed using the square root of the reflectance to achieve a normal distribution. 

#### 2.4.2. Statistics

Differences between EC and non-EC parents, EC tertiles, and median based-categories of parent outcome measures were assessed with t-test, analysis of variance (ANOVA), Mann–Whitney *U* (continuous variables), and chi square (categorical variables). A Scheffe test was used to examine unplanned comparisons among group means in ANOVA. Study power to detect a difference of either four points on self-efficacy scale, two points on modeling scale, or two types of FV between EC and non-EC parents was 0.9. 

Cluster analyses were employed to identify relatively homogeneous subgroups within the sample. Independence among variables (SE/OE, modeling, vegetable availability, and fruit availability) was confirmed (*r* ≤ 0.32) before being entered into the analysis. Two-step cluster analysis used sequential and hierarchical agglomerative methods pre-clustering and then sub-clustering of the data. The log-likelihood measure was used as a distance measure. The number of clusters was determined based on the largest relative increase in distance between the two closest clusters defined by the Schwartz Bayesian criterion [44]. All clustering variables were standardized and a listwise deletion approach to missing data was used. *t*-tests and chi-square tests were used to test for differences between clusters (independent variable) on parent demographic characteristics, attitudes, and behaviors. 

In addition, differences in parent eating competence status across cluster groups were assessed using a general linear model that included parent body mass index (BMI) as a covariate and income status as a cofactor. General linear model results are reported as estimated marginal means ± standard errors. Level of significance was set at *p* < 0.05. 

## 3. Results

Of the 564 that accessed the online survey, 450 consented and 424 completed the survey. Survey completers were nearly all non-Hispanic white females with some post-high school education (Table 1) and a mean age of 39.1 ± 6.0 years. Hispanic and non-Hispanic parents did not differ on proportion that were EC or scores on the ecSI 2.0^TM^, modeling, SE/OE, and FV availability measures. Non-Hispanic ethnicity was associated with a higher education level (*p* = 0.001). Parents not completing the ecSI 2.0^TM^ items in the survey set (*n* = 16) did not differ from ecSI 2.0^TM^ completers on any demographic or primary outcome measures. Of the 408 parents completing all ecSI 2.0^TM^ items, 53% (*n* = 214) were EC. 

### 3.1. Relationship between Eating Competence and Demographics 

Mean ecSI 2.0^TM^ scores and subscale scores did not differ significantly between males (32.5 ± 8.5, *n* = 55) and females (32.1 ± 7.9, *n* = 350). Parents denoted as low-income had significantly lower ecSI 2.0^TM^ scores compared with other parents (30.3 ± 8.3, *n* = 145; 33.1 ± 7.7, *n* = 263, respectively; *p* = 0.001). BMI was lower (*p* < 0.001) in EC (*n* = 212) than non-EC parents (*n* = 190) (24.5 ± 4.9 vs. 27.7 ± 6.2). Also, a lower proportion of EC parents were overweight or obese compared with those not EC (11% vs. 27%, respectively; *p* < 0.001). 

### 3.2. Comparisons between EC and Non-EC Parents 

EC parents reported more frequent modeling behaviors related to meals and FV intake compared with non-EC parents (16.1 ± 4.2 and 14.2 ± 4.1, respectively; *p* < 0.001). In addition, EC parents more frequently ate dinner, vegetables at dinner, vegetables at lunch, fruit as a snack, and fruit at breakfast with their child (Figure 1). 

EC parents averaged higher SE/OE scores related to FV preparation and consumption compared with non-EC parents (*p* < 0.001). Differences were statistically significant (all *p* ≤ 0.002) for each of the 12 SE/OE survey items (Table 2). Vegetables were more available in the homes of EC parents compared with non-EC parents (6.8 ± 1.5 vs. 6.3 ± 1.8, respectively, *p* = 0.002), and combined FV availability was also higher in the homes of EC parents (12.4 ± 2.6 vs. 11.6 ± 2.8, respectively, *p* = 0.002). Fruit or juice availability did not differ according to EC status (*p* = 0.08 vs. *p* = 0.24, respectively). After controlling for education level, differences between EC and non-EC parents remained significant for the modeling (*p* = 0.001), in-home FV availability (*p* = 0.006), and SE/OE measures (*p* = 0.015).

### 3.3. Comparisons among Parents according to Eating Competence Tertile 

SE/OE, parent modeling, and in-home vegetable availability also differed among ecSI/LI tertiles (data not shown, all *p* ≤ 0.007). Post hoc Scheffe tests for these analyses revealed that parents in the lowest ecSI 2.0^TM^ tertile had significantly lower modeling scores than parents in the high and middle ecSI 2.0^TM^ tertiles, and in-home vegetable availability was greater in the high than in the low tertile. SE/OE scores were significantly greater for parents in the high ecSI 2.0^TM^ tertile than in the middle or lower tertiles. Parent education was associated with tertile of ecSI 2.0^TM^ score (*p* = 0.004). The proportion of parents who attended graduate school was highest among those in the high tertile (38% vs. 20% in the low and 33% in the middle tertiles). After controlling for education level, differences among tertiles remained significant for modeling and in-home vegetable availability (*p* < 0.02), with a trend toward difference (*p* = 0.08) for SE/OE.

### 3.4. Median Analyses

ecSI 2.0^TM^ scores were significantly higher for parents at or above the median for modeling, vegetable availability, and SE/OE (all *p* < 0.01) scores. All 4 ecSI 2.0^TM^ subscales were significantly higher for parents at or above the median modeling scores. Eating attitudes, food acceptance, and contextual skills subscale scores were higher for parents at or above the median vegetable availability score. Food acceptance, contextual skills, and internal regulation subscale scores were higher for parent at or above the median SE/OE scores (Table 3).

### 3.5. Cluster Analyses

The cluster analysis of parental food and eating behaviors delineated two clusters depicting *thrivers* (*n* = 255) and *strivers* (*n* = 151). Compared to *strivers, thrivers* had more self-efficacy for providing fruits and vegetables to their child(ren), modeled more healthful eating behaviors, and identified higher in-home fruit and vegetable availability. Clusters, which did not differ in age, gender, BMI, or income level, were validated with significantly higher ecSI 2.0^TM^ (*p* < 0.001) and subscale scores (all *p* ≤ 0.003) in *thrivers* compared with *strivers* (Table 4). Compared to *thrivers,* more *strivers* used an authoritarian parenting style (23% vs. 40%, *p* < 0.001) and had lower levels of physical activity (18% vs. 32%, *p* = 0.004). Eating competence scores (ecSI 2.0^TM^) remained greater (*p* < 0.001) among *thrivers* (33.5 ± 0.5) than in *strivers* (29.5 ± 0.6) when BMI, income level, or education were included in the general linear model. The average silhouette was of sufficient quality at 0.4. The discriminant function as a whole was significant (Wilks lambda = 0.32; *p* < 0.001). SE/OE and vegetable availability were the strongest predictors for cluster delineation. Derivation variables are shown in Table 4.

## 4. Discussion

### 4.1. Summary

The results from this relatively large sample of non-Hispanic white parents of 4th grade youth supported the tenets of EC [4]. Eating and feeding behaviors, physical activity level, weight status, and demographic characteristics were assessed and compared according to EC status. EC parents demonstrated stronger role modeling behaviors (e.g., eating dinner, eating vegetables at dinner, and cooking with their child). Parent behaviors clustered into two groups (*thrivers* and *strivers*) that did not differ in age or gender. However, *thrivers* not only demonstrated more role modeling, SE/OE behaviors, and home FV availability, but they were more physically active, less likely to utilize an authoritarian parent feeding style and more EC even after adjusting for BMI and income. These findings corroborated those from the mostly Hispanic sample [5] with the exception that unlike the Hispanic sample, which had significantly different ecSI 2.0^TM^ scores between those below or at and above the median for home FV availability (32.4 ± 8.2 vs. 34.8 ± 8.7; *p* < 0.05) [5], ecSI 2.0^TM^ scores in the current study were not significantly different (31.2 ± 8.3 vs. 32.7 ± 7.8). The comparative findings are remarkable given that the two samples differed in ethnicity (white ethnicity reported for 14% in the previous sample and 88% in this sample) and education level (4-year college degree or higher for 18% vs. 65%; high school or less for 51% vs. <7%). Both samples were otherwise similar; for example, they were mostly female (89% vs. 86%), with a mean ecSI score >32 (33.6 ± 8.5 vs. 32.1 ± 8.0) and similar modeling of behaviors related to meals and FV (15.3 ± 4.8 vs. 15.2 ± 4.2).

### 4.2. Consequences of Reproducibility

Replication has been construed as either “conceptual” or “direct”. Conceptual replication tries to validate and further define the underlying theory, whereas direct replication involves validating a specific fact or result. Direct replication studies retain all features of the original study, but conceptual replication intentionally varies one or more features to examine generalizability across conditions [30,31,32,45]. Earp and Trafimow [30] note that the distinction between the two types is not absolute; and so it is with this study. We sought to determine if the findings relating EC and parent modeling behaviors were reproduced, but intentionally evaluated for this in a sample that differed in ethnicity from the original study, that is, conceptual replication. Usually, little support is given for this type of endeavor, especially for publication and promotion credit [31,45], and the consequences of failure to replicate challenge this undertaking. 

However, as noted in a seminal paper published in Science as an Open Science Collaboration [46], “Innovation points out paths that are possible; replication points out paths that are likely; progress relies on both. Replication can increase certainty when findings are reproduced and promote innovation when they are not.” Findings from this study point out a likely path that being EC is a desired state for either white or Hispanic parents’ own health as well as their child’s. They also encourage ascribing the findings to other races, ethnicities, cultures, and countries, especially the non-English speaking cultures. However, the results also show that affirming replication in said groups is feasible, reassuring, and will enhance the study of the EC construct, furthering the development of strategies, education, and policies that can promote healthful eating behaviors through the adoption of an EC approach.

### 4.3. Strengths and Limitations

A strength of this study was that the sample was similar to the previously studied, mostly Hispanic group, facilitating generalization of findings from Hispanic to non-Hispanic white parents. Both samples were parents of 4th grade youth, similar in age; sex; frequency of eating dinner with their child; ecSI 2.0^TM^ and subscale scores; and the primary outcomes of modeling, SE/OE, and FV availability in the home. As a result, this study was a replication across groups differing in proportion of Hispanic ethnicity and related level of education; the mostly Hispanic sample included 49% with at least some college post-secondary education, compared with 94% for this mostly non-Hispanic white sample. Both studies utilized identical, validated instruments, although data collection was via paper survey in the Hispanic sample rather than the online survey platform used in the current study. In addition to the strength of sample similarity, the current study also included measures, for example, self-reported weight, that were noted as missing factors in the earlier study, enabling further exploration of relationships. For example, this study allowed for comparisons across clusters that controlled for BMI and income status, thus strengthening the earlier findings with Hispanic parents. 

Findings from this study increased the certainty that EC parents are better aligned with many child obesity prevention practices. However, limitations or threats to reproducibility of findings were still apparent. Although participants were from two school districts with children in eight elementary schools, generalization is attenuated because participants were from Fort Collins and Loveland, Colorado, which is a uniquely healthy state. It has the United States’ lowest rate of adult obesity, second highest level of physically active adults, and is eighth highest in the number of adults eating five or more daily FV servings. Coloradan children also stand out, being ranked nationally with the fifth lowest rate of obesity and above the national average for physical activity, with 67% of school-age children participating in vigorous physical activity four or more days each week. In addition, at the time of this study, Fort Collins had just implemented a policy removing unhealthy beverages from vending machines in all city buildings and several hospitals across Colorado joined this action, revealing a strong health promotion culture [20]. Findings are also limited because height, weight, and income-based assistance program use was self-reported, possibly misrepresenting these sample characteristics and EC relationships for comparison with future studies.

## 5. Conclusions

Parental supports, for example, eating meals together, serving FV as snacks, encouraging breakfast, and facilitating an active lifestyle, impact child health-related behaviors and outcomes [47]. EC is associated with more frequent role modeling of healthful eating practices, self-efficacy to serve FV, and home availability of FV for Hispanic and non-Hispanic white parents of 4th grade youth. Efforts to implement parent-driven childhood obesity prevention strategies are informed by parent EC status. These findings argue for increased attention to the development of EC through education [48], policy, system and environmental supports. 

## Figures and Tables

**Figure 1 nutrients-11-01501-f001:**
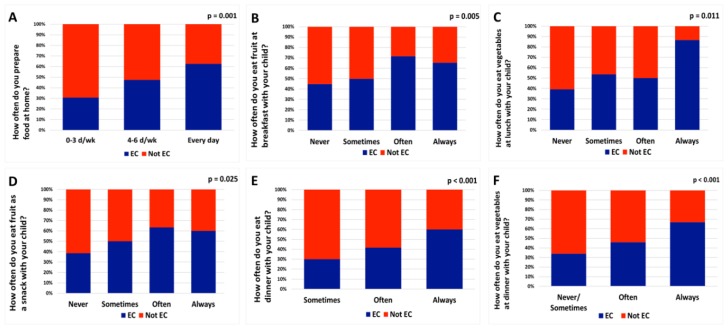
Parent eating competence (EC) proportion mapped to response options for selected modeling items. Response option of selected modeling survey item by EC and non-EC parents of 4th grade children. Figure 1 shows percent of total responses. Described below for each panel are the total number of parents that responded, the total number of parents for the response option, and the total number of EC parents for the response option. (**A**) Total responses, *n* = 408; 0–3 days/week (total *n* = 26, EC *n* = 8); 4–6 days/week (total *n* = 219, EC *n* = 104); every day (total *n* = 163, EC *n* = 102). (**B**) Total responses, *n* = 408; never (total *n* = 103, EC *n* = 46); sometimes (total *n* = 223, EC *n* = 111); often (total *n* = 56, EC *n* = 40); always (total *n* = 26, EC *n* = 17). (**C**) Total responses, *n*
*=* 407; never (total *n* = 56, EC *n* = 22); sometimes (total *n*
*=* 308, EC *n* = 165); often (total *n* = 28, EC *n* = 14); always (total *n* = 15, EC *n* = 13). (**D**) Total responses, *n* = 407; never (total *n* = 44, EC *n* = 17); sometimes (total *n* = 244, EC *n* = 122); often (total *n* = 104, EC *n* = 66); always (total *n* = 15, EC *n* = 9). (**E**) Total responses, *n* = 407; sometimes (total *n* = 20, EC *n* = 6); often (total *n* = 132, EC *n* = 55), always (total *n* = 255, EC *n* = 153). (**F**) Total responses, *n* = 407; never/sometimes (total *n* = 67, EC *n* = 23); often (total *n* = 172, EC *n* = 79); always (total *n* = 168, EC *n* = 112). EC, eating competence. *P* values denote a test of whether or not response options for each question are independent of eating competence status as measured by Chi-square.

**Table 1 nutrients-11-01501-t001:** Characteristics of parents of 4th grade participants in *Fuel for Fun*
^1^.

Variable	*n* (%)
Gender	
Female	362 (86)
Race	
White	372 (88)
American Indian/Alaska Native	7 (2)
Black	3 (<1)
Asian	15 (4)
Native Hawaiian/Pacific Islander	2 (<1)
Ethnicity	
Hispanic, White	32 (8)
Hispanic, Black	1 (<1)
Education	
Did not complete high school	2 (<1)
High school graduate or GED	25 (6)
Some college/2y college degree	123 (29)
4y college degree	147 (35)
Post-graduate college	127 (30)
Parent feeding style	
Uninvolved	80 (20)
Indulgent	122 (29)
Authoritarian	121 (29)
Authoritative	89 (22)
BMI Category	
Underweight	6 (1)
Normal weight	211 (50)
Overweight	121 (29)
Obese	80 (19)
IPAQ Category	
Low	97 (23)
Moderate	122 (29)
High	205 (48)
Self-identify as physically activeYes	292 (69)
Low-income ^2^	152 (36)
ecSI 2.0 score^TM 3^	32.1 ± 8.0
Subscales	
Eating attitudes	10.6 ± 2.9
Food acceptance	5.1 ± 2.1
Internal regulation	6.3 ± 2.0
Contextual skills	10.2 ± 3.0
SE/OE score ^4^	55.0 95% CI 54.3–55.6
Modeling ^5^	15.2 ± 4.2
In-home fruit availability ^6^	4.4 ± 1.4
In-home vegetable availability ^7^	6.6 ± 1.7
In-home juice available ^8^	1.1 ± 0.9
Total in-home fruits/veg available ^9^	12.0 ± 2.7
Parent perceived stress ^10^	6.7 ± 2.1

GED, General Education Diploma; BMI, body mass index; IPAQ International Physical Activity Questionnaire; ecSI 2.0^TM^, Satter eating competence inventory; FV, fruits and vegetables; SE/OE, self-efficacy/outcome expectancies for fruits and vegetables. ^1^ Values are *n* (%) or mean ± SD, total *n* = 424 although n varies throughout the tables because of missing or incomplete responses ranging from 408 to 422 when not 424. % may not add to 100 because of rounding. Race/ethnicity counts are affirmed responses and reported as a percent of the total sample. ^2^ Income-based assistance program participant or always, often worries about money for food. ^3^ ecSI 2.0 ^TM^ possible score 0–48; possible subscale scores: eating attitudes and contextual skills, 0–15; food acceptance and internal regulation, 0–9. ^4^ SE/OE survey; possible score 12–60. SE/OE summed score transformed to achieve normal distribution and are reported as means and 95% confidence intervals. Table entries are back transformed values. ^5^ Survey of modeling mealtime and FV-related behaviors; possible score 0–33. ^6^ Number of fruits available in the home; possible range 0–8. ^7^ Number of vegetables available in the home; possible range 0–9. ^8^ Number of juices available in the home; possible range 0–3. ^9^ Fruits, vegetables, and fruit juices summed; possible range 0–20. ^10^ Visual analog scale from 0 (no stress) to 10 (extreme stress).

**Table 2 nutrients-11-01501-t002:** Self-efficacy/outcome expectancies (SE/OE) score and item responses compared between EC and non-EC parents ^1^.

	EC/non-EC	EC Parents ^2^	Non-EC Parents ^3^
	*n*/*n*		
Total SE/OE score ^4^	212/189	56.3 55.6–57.0 ***	53.2 52.0–54.4
Survey items ^5^			
I can prepare fruit that my child will eat	214/194	4.8 ± 0.7 ***	4.5 ± 1.0
I can prepare vegetables that my child will eat	213/194	4.7 ± 0.7 ***	4.3 ± 1.1
I can prepare fruit that my child will like	213/194	4.8 ± 0.6 ***	4.5 ± 0.9
I can prepare vegetables that my child will like	214/194	4.6 ± 0.8 ***	4.2 ± 1.1
I can prepare a recipe with my child	213/192	4.6 ± 0.7 ***	4.3 ± 1.0
If I buy fruit, my child will eat it	214/192	4.6 ± 0.7 ***	4.4 ± 1.0
If I buy vegetables, my child will eat them	214/194	4.4 ± 0.9 ***	4.0 ± 1.1
If I give my child fruit for a snack, my child will eat the fruit	214/193	4.6 ± 0.8 ***	4.3 ± 1.0
If I give my child vegetables for a snack, my child will eat the vegetables	214/193	4.1 ± 1.1 **	3.7 ± 1.3
If I include fruit as part of a meal, my child will eat the fruit	214/194	4.7 ± 0.8 ***	4.4 ± 0.9
If I include vegetables as part of a meal, my child will eat the vegetables	214/194	4.4 ± 0.9 ***	4.0 ± 1.1
If I prepare a meal together with my child, my child will eat the meal	214/193	4.5 ± 0.8 ***	4.2 ± 0.9

^1^ Values are means ± SD, except for Total SE/OE score, which is mean and confidence intervals. Asterisks indicate different from non-EC parents: ** *p* ≤ 0.01, *** *p* ≤ 0.001. EC, eating competent; SE/OE, self-efficacy/outcome expectancies; ecSI 2.0^TM^, Satter Eating Competence Inventory 2.0. ^2^ Defined by ecSI 2.0 ^TM^ score ≥ 32. ^3^ Defined by ecSI 2.0 ^TM^ score < 32. ^4^ Possible scores 12 (low SE/OE) to 60 (high (SE/OE). SE/OE summed scores transformed to achieve normal distribution and reported as mean and 95% confidence intervals. Table values are back transformed and may exceed 60. ^5^ For each survey item, possible response is 1 (strongly disagree) to 5 (strongly agree). EC and non-EC compared with Mann–Whitney *U*.

**Table 3 nutrients-11-01501-t003:** ecSI 2.0^TM^ scores compared between parents below and at/above median survey scores ^1^.

	SE/OE ^2^		Modeling ^3^		In-Home FV Availability ^4^		In-Home Fruit Availability ^5^		In-Home Vegetable Availability ^6^	
Eating Competence ^7^	<Median	≥Median	<Median	≥Median	<Median	≥Median	<Median	≥Median	<Median	≥ Median
ecSI 2.0^TM^ score	30.5 ± 7.6 *** *n* = 180	33.6 ± 8.0 *n* = 221	29.5 ± 8.1 *** *n* = 169	34.1 ± 7.3 *n* = 232	31.2 ± 8.3 *n* = 168	32.7 ± 7.8 *n* = 240	32.0 ± 8.7 *n* = 92	32.1 ± 7.8 *n* = 316	30.7 ± 8.4 ** *n* = 172	33.2 ± 7.6 *n* = 236
ecSI 2.0^TM^ subscales										
Eating attitudes	10.3 ± 2.7 *n* = 183	10.9 ± 3.1 *n* = 227	10.1 ± 3.2 *** *n* = 172	11.0 ± 2.6 *n* = 236	10.4 ± 3.0 *n* = 173	10.8 ± 2.8 *n* = 245	10.6 ± 3.3 *n* = 96	10.6 ± 2.8 *n* = 322	10.3 ± 3.1 * *n* = 178	10.9 ± 2.8 *n* = 240

Food acceptance	4.6 ± 2.0 *** *n* = 185	5.6 ± 1.9 *n* = 226	4.5 ± 2.0 *** *n* = 174	5.6 ± 1.9 *n* = 236	4.9 ± 2.1 * *n* = 172	5.3 ± 2.0 *n* = 246	4.9 ± 2.2 *n* = 95	5.2 ± 2.0 *n* = 323	4.8 ± 2.1 ** *n* = 179	5.4 ± 2.0 *n* = 239

Internal regulation	5.9 ± 2.0 *** *n* = 186	6.6 ± 2.0 *n* = 227	6.0 ± 2.2 * *n* = 174	6.4 ± 1.9 *n* = 238	6.2 ± 2.1 *n* = 174	6.3 ± 2.0 *n* = 247	6.5 ± 2.2 *n* = 97	6.2 ± 2.0 *n* = 324	6.1 ± 2.1 *n* = 180	6.4 ± 2.0 *n* = 241

Contextual skills	9.6 ± 2.9 *** *n* = 185	10.7 ± 2.9 *n* = 228	9.0 ± 2.9 *** *n* = 173	11.1 ± 2.7 *n* = 237	9.8 ± 3.1 *n* = 173	10.4 ± 2.9 *n* = 247	10.1 ± 3.2 *n* = 96	10.2 ± 2.9 *n* = 324	9.6 ± 3.2 *** *n* = 180	10.6 ± 2.7 *n* = 240

^1^ Values are means ± SD. Asterisks indicate different from ≥ median: * *p* < 0.05, ** *p* ≤ 0.01, *** *p* ≤ 0.001. Eating competent, EC; Satter Eating Competence Inventory for Low-Income, ecSI 2.0 ^TM^; self-efficacy/outcome expectancies, SE/OE; fruit and vegetables, FV. ^2^ SE/OE, possible score 12–60; median = 56. ^3^ Modeling survey, possible score 0–33; median = 15. ^4^ In-home fruit and vegetable availability, possible range 0–20; median = 12. ^5^ In-home fruit availability, possible range 0–8; median = 4. ^6^ In-home vegetable availability, possible range 0–9; median = 7. ^7^ ecSI 2.0^TM^, possible score 0–48; possible subscale scores: Eating attitudes and contextual skills 0–15, food acceptance and internal regulation 0–9.

**Table 4 nutrients-11-01501-t004:** Behaviors compared between parents in clusters of thriving or striving for healthful and competent feeding behaviors ^1^.

Measure	Thriver/Striver	Thrivers	Strivers
	*n*/*n*	Mean/95% CI	Mean/95% CI
**Derivative Variables**			
SE/OE score ^2^	255/151	58.4/58.1–58.6	46.1/44.7–47.5
		Mean ± SD	
Modeling score ^3^	255/151	16.5 ± 3.8	12.8 ± 3.9
Fruit Availability ^4^	255/151	4.6 ± 1.2	3.8 ± 1.5
Vegetables Availability ^5^	255/151	7.2 ± 1.2	5.6 ± 1.8
**Validation Variables**			
ecSI 2.0^TM^ score ^6^	250/144	33.7 ± 7.6 ***	29.6 ± 7.8
ecSI 2.0^TM^ Subscales			
Eating attitudes	253/147	11.0 ± 2.8 **	10.1 ± 2.9
Food acceptance	254/149	5.6 ± 1.9 ***	4.3 ± 2.0
Internal regulation	254/150	6.5 ± 2.0 ***	5.9 ± 2.1
Contextual skills	254/149	10.7 ± 2.8 ***	9.3 ± 3.0
BMI	250/150	25.7 ± 5.9	26.5 ± 5.7
Age (years)	254/150	38.8 ± 5.9	39.7 ± 6.0
		Frequency *n* (%)	
Parent Feeding Style ***			
Uninvolved		49 (20)	29 (20)
Indulgent		91 (36)	28 (19)
Authoritarian		58 (23)	58 (40)
Authoritative		53 (21)	30 (21)
Gender			
Male		32 (13)	23 (15)
Female		221 (87)	127 (85)
IPAQ Category ^7,^**			
Low		46 (18)	49 (32)
Moderate		76 (30)	40 (26)
High		133 (52)	62 (41)
Low-Income Status			
Low-income ^8^		89 (35)	56 (37)
Not low-income		166 (65)	95 (63)

^1^ Values are *n* (%) or mean ± SD, total *n* = 406, although *n* varies throughout the tables because of missing or incomplete responses ranging from 394 to 404 when not 406. Asterisks indicate different from striver parents: ** *p* ≤ 0.01, *** *p* ≤ 0.001. SE/OE, self-efficacy/outcome expectancies; ecSI 2.0^TM^, Satter eating competence inventory. ^2^ SE/OE possible score 12–60; SE/OE summed scores transformed to achieve normal distribution and may exceed 60. Table entries are back transformed. ^3^ Modeling survey possible score 0–33. ^4^ Number of fruits available in the home possible range 0–8. ^5^ Number of vegetables available in the home, possible range 0–9. ^6^ ecSI 2.0^TM^ possible score 0–48; possible subscale scores: eating attitudes and contextual skills, 0–15; food acceptance and internal regulation, 0–9. ^7^ IPAQ, International Physical Activity Questionnaire; responses converted to met min/week and identified as low, moderate, and vigorous activity categories. ^8^ Parents who reported often or always worrying about money or participation in an income-based program, for example, the Supplemental Nutrition Assistance Program or Women, Infants, and Children Supplemental Nutrition Program, were defined as low income.

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
