# Peer review of "Eating Competent Parents of 4th Grade Youth from a Predominantly Non-Hispanic White Sample Demonstrate More Healthful Eating Behaviors than Non-Eating Competent Parents"

_nutrients, 2019, doi:10.3390/nu11071501_

Round 1
Reviewer 1 Report
In this manuscript, the authors described various characteristics comparing eating competent (EC) and non-EC parents in a non-Hispanic sample and relationships between parent eating and feeding behaviors and EC. The authors replicated the findings previously reported in a Hispanic sample.
Minor comments:
Line 90: Authors mention that parents clicked a box on the website after reading the consent to indicate their agreement. Were study personnel available to answer any questions that the potential participants might have? A statement mentioning this should be included.
Line 100: extra space before "11"
Table 1: How relevant is perceived stress to this manuscript. Does it add anything to the narration? The authors do not mention anything about it in the results or discussion.
Author Response
Reviewer Comment | Authors’ Response |
Reviewer 1 | |
In this manuscript, the authors described the various characteristics comparing eating competent (EC) and non-EC parents in a non-Hispanic sample and relationships between parent eating and feeding behaviors and EC. The authors replicated the findings previously reported in a Hispanic sample. | This is a correct and succinct summary. Thank you for your review. |
Minor Comments: Line 90: Authors mention that parents clicked a box on the website after reading the consent to indicate their agreement. Were study personnel available to answer any questions that the potential participants might have? A statement mentioning this should be included. | The consent form included an email and phone number to contact with questions and, as noted in line 90, they were able to print the consent form for later reference. Availability of personnel to answer questions was added to line 90. |
Line 100: extra space before “11” | This is because of the journal template requiring a full justification of text. |
Table 1: How relevant is perceived stress to this manuscript. Does it add anything to the narration? The authors do not mention anything about it in the results or discussion. | We mention the mean level of stress in the last line of Table 1 in the Results. This provides additional information about our sample. Stress has been shown in other studies to be a factor in people’s food behaviors, which is why we added it to our survey set. The findings show that our sample had some level of stress, but not extreme. We limited our discussion of stress to indicating the level of participants to avoid detracting from the main purpose of the study. If you prefer, we can remove this piece of information about our sample. |

Reviewer 2 Report
This paper was interesting to read. This was a cross-sectional study that surveyed parents of 4th graders on many different measures. Please consider the following comments and suggestions:
General comment: Your objectives are to compare the outcomes of a survey from a previous study done in Hispanic parents but you did not showcase any results that compares the two populations (there were no comparisons just the findings from the non-Hispanic population). I suggest you include some direct comparisons as this was the primary objective.
Throughout the text there were some spacing issues. I am not sure if it’s the Editor’s side or yours, please check.
Intro:
Line 40: In your description of competent eaters, you talk about “amounts that are satisfying”… I am not sure if this is referring to them eating foods that are “high-fat, high sugar”, in other words their ability to control their intakes of “unhealthier food choices”? Please clarify.
Your description of the importance to reproduce the study by quoting Schmidit is in my opinion not necessary. You can simply state in general words (paraphrase) and reference, no need to quote. When reviewing your introduction after the abstract, I would have never questioned your objectives. By putting this paragraph in, it made me wonder what the validity of the study was with this justification. In other words, I felt this was short-changing yourselves.
M&M
What is meant by “impact” in 83 classrooms? Sentence is off…
PA : it was only after reviewing your tables that I understood you were surveying parents PA levels not child. Please add this to the 2.3.5 paragraph.
A comment: I would have rather seen the comparisons side-to-side from the previous study instead of Table 3. The tertiles are not as interesting and it would be of more value to see what the parents from Study 1 reported compared to this study. I suggest removing Table 3.
Did you survey on number of children/ family members per household? What about seasonality effects? Which season was this administrated?
Tables/ Figures
Please justify your tables to have your left columns to the left- very hard to read.
Table 1: Can you put the PA measures together. You have listed the PA yes/ no then lower down review the IPAQ results. This table is filled with rich information and I found it so hard to follow.
Figure 1 needs a 1. I suggest you change the format of this figure to box plots so you can clearly ID the significance because as it stands I have no idea what the p-values are referring too. Also, add the n in each or on top of each box-plot to reduce the massive paragraph under the Figure. In other words, amazing data but make it super simple for the reader please.
Table 2 and 4: justification issues
Author Response
Reviewer Comment | Authors’ Response |
Reviewer 2 | |
This paper was interesting to read. This was a cross-sectional study that surveyed parents of 4th graders on many different measures. Please consider the following comments and suggestions: | Thank you for your review. |
General comment: Your objectives are to compare the outcomes of a survey from a previous study done in Hispanic parents but you did not showcase any results that compares the two populations (there were no comparisons just the findings from the non-Hispanic population). I suggest you include some direct comparisons as this was the primary objective. | Our objective was to determine if the findings from the previous study with Hispanic parents were replicable in a sample that was not predominantly Hispanic. The summary [starting line 303] reports that all but one finding was replicated, therefore it didn’t seem a good use of space to list all the similar findings between the two samples. However, we have added in the details for the results that did differ between the two samples. {lines 314-315). In addition, we have added into the summary some background information comparing some demographics of the two samples that also serves to emphasize the value of the generalizability of the conclusions. Please see lines 318-323: “The comparative findings are remarkable given that the two samples differed in ethnicity (white ethnicity reported for 14% in the previous sample and 88% in this sample) and education level (4-year college degree or higher for 18% vs. 65%; high school or less for 51% vs. < 7%). Both samples were otherwise similar; for example, they were mostly female ( 89% vs. 86%), with a mean ecSI score > 32 (33.6 ± 8.5 vs. 32.1 ± 8.0) and similar modeling of behaviors related to meals and FV (15.3± 4.8 vs. 15.2 ± 4.2).” |
Throughout the text there were some spacing issues. I am not sure if it’s the Editor’s side or yours, please check. | This is a result of the manuscript template. That requires full justification. I have contacted the editor who assured me that spacing issues will be addressed prior to publication. |
Intro: Line 40: In your description of competent eaters, you talk about “amounts that are satisfying”… I am not sure if this is referring to them eating foods that are “high-fat, high sugar”, in other words their ability to control their intakes of “unhealthier food choices”? Please clarify. | Eating competence does not focus on specific foods or food components, but rather on persons being comfortable with their eating, i.e., what they eat, how much, when, where, but consider dietary variety, eating what is good for them, planning for feeding, eating regular meals. As a result, dietary adequacy, and impact on body weight are usually questioned when one first encounters eating competence. We have conducted several studies that have shown that persons who had the discipline to plan for eating but didn’t strictly control what or how much they ate, actually had better dietary quality [e.g., references 7, 9] and more normal BMI. In other words eating competence seems to be a proxy for being reasonable, i.e., eating regular meals, including a variety of foods, but not getting hung up on low-fat, low-carb, plant-based only etc….they eat in a way that fits with their lifestyle but care about eating well for them. Does this help? If you prefer, I can share a copy of many of the references cited in the Introduction that are about Eating Competence, especially Reference 4. |
Your description of the importance to reproduce the study by quoting Schmidt is in my opinion not necessary. Yu can simply state in general words (paraphrase) and reference, no need to quote. When reviewing your introduction after the abstract I would have never questioned your objectives. By putting this paragraph in, it made me wonder what the validity of the study was with the justification. In other words, I felt this was short-changing yourselves. | We removed the quote as you suggested; it was originally included to strengthen the statement that the call to attend to replication is urgent in social sciences. That statement had also included the Schmidt reference. |
M&M What is meant by “impact” in 83 classrooms? Sentence is off… | The study from which these baseline data were utilized was a controlled, randomized study to the results of the primary study have the level of denoting impact, rather than association. To clarify we have listed the specific impact outcomes on line 80, “. . . impact on FV preferences, attitudes and self-efficacy toward cooking and making foods. . .” |
PA: It was only after reviewing your tables that I understood you were surveying parents PA levels, not child. Please add this to the 2.3.5 paragraph. | Thank you for this clarification. So noted on line 126. |
A comment: I would have rather seen the comparisons side-to side from the previous study instead of Table 3. The tertiles are not as interesting and it would be of more value to see what the parents from Study 1 reported compared to this study. I suggest removing Table 3. | We included tertile analyses and median analyses (as shown in Table 3) because these results were included in the examination of the Hispanic sample. Unless there is strong opposition, we would like to keep them in this paper to effect as much comparison between the two samples as possible. We agree that describing the comparisons between the two samples would be useful, especially for those who do not have access to the Journal of Nutrition article. With the exception of the item on lines 312-315 all findings were comparable suggesting a better use for manuscript space. Please see above for a response about how we added some comparisons between the two studies. |
Did you survey on number of children/family members per household? What about seasonality effects. Which season was this administered? | Yes, we asked how many children were in these age groups: < 2; 2-5; 6- 10; 11-18; > 18 and then we summed the responses to know total number of children in the family. We did not ask about family size (e.g., extended family members). The mean number of children/family was 2.4 ± 1.1. Here is the information in more detail: 1 child 16%; 2 children 50%; 3 children 21%; 4 children 9%; 5 children 1%; 6 children 2%; 7 or more children 1% . Breakdown by age is as follows: [e.g., 26 had 1 child < 2 y; 176 had 2 children 6 – 10 y old] # youth< 2 y2– 5 y6–10 y11 –18 y>18 y12686240118182181763853 120514 22 5 1
Since the number of children/family was not unusual, we did not include this information in the sample description; however if you want this included certainly, we can do so.
You are correct to be concerned about differences based on seasonality of data collection. However, seasonality difference is not an issue in this report because these data were all collected in mid-September/mid-October after the school year had started. |
Tables/Figures Please justify your tables to have your left columns to the left-very hard to read. | I agree but the table was developed with the template provided by the journal. My understanding is that the columns will be made more legible when edited for publication. To help at this stage, I’ve double spaced between rows when possible to separate categories of findings/descriptions for easier reading. |
Table 1: Can you put the PA measures together. You have listed the PA yes/no then lower down review the IPAQ results. This table is filled with rich information and I found it so hard to follow. | Yes, this was done. Thanks for the suggestion. |
Figure 1 needs a 1. I suggest you change the format of this figure to box-plots so you can clearly ID the significance because as it stands I have no idea what the p-values are referring too. Also, add the n in each or on top of each box-plot to reduce the massive paragraph under the Figure. In other words, amazing data but make it super simple for the reader please. | A “1” was added when denoting the figure. This is the format used to show these data for the Hispanic sample published earlier so we want to replicate the presentation of these findings. Each bar represents a stacked graph of EC vs non-EC. In our previous paper we had the n’s in the graph, but several reviewers asked us to remove them to a caption because they made the graph too busy. We agree that the meaning of P should be clearer. The P values represent the significance of the test of association between eating competence and response options for each of the questions posed as measured by Chi Square. Therefore we have added to the caption on line 231: P values denote a test of whether or not response options for each question are independent of eating competence status as measured by Chi Square. |
Tables 2 and 4: justification issues | As noted above, the information was inserted into the manuscript submission template. I queried the editor about this concern and was told that it would be fixed in final preparation of the manuscript. To help at this stage, I’ve double spaced between rows when possible to separate categories of findings/descriptions for easier reading. |

Round 2
Reviewer 2 Report
Thank you for the revisions and your comments. I appreciate the spacing issues now: as a reviewer it was hard to not notice but good to know beyond your control.
A few more things to consider as I review your interesting manuscript again:
Abstract conclusion: Can you edit it? You mention nutrition education in youth but this survey is in the parents.
Intro: Line 38 you are missing a ")" at the end of your e.g.,
Intro: remove the comma after "provide" nourishing...
Intro: Line 41: Can you make this sentence more clear ... they are comfortable with their enjoyment... what does this mean? This can be made more clear for the reader.
Intro line 43: Is there where you are now reviewing the Hispanic sample? If yes, then make this super clear! I really missed this connection the last time. Maybe a new paragraph to really highlight your findings from the previous cohort
Intro: final paragraph. Thank you for editing. As you stated, reproducibility is not new, so I really do not feel you need to justify your manuscript in this way. Why don’t you just make this paragraph a quick summary of your Hispanic study results above and say we wanted to expand the population to a new cohort to see if the results are the same?
M&M
You doubled up on the word “impact” when you edited the edits
Discussion:
Your edits “analyses” does not work in this sentence. Please change.
Once again, in your discussion for 4.2, I do not think it is necessary to quote others on why reproducibility and replication of studies is important. It is part of the research process and scientific discovery. I would paraphrase this and just add your references. You do not need to justify your study in this manner: your study is sound and you have shown nice results.
I really appreciate your edits to add more comparisons between cohorts. Thank you for considering my suggestion.
